# Learning to Scale Logits
# for Temperature-Conditional GFlowNets

**Minsu Kim**[*], **Joohwan Ko**[*]
KAIST

**Dinghuai Zhang, Ling Pan**
Mila, Université de Montréal

**Taeyoung Yun, Woochang Kim, Jinkyoo Park**
KAIST

**Yoshua Bengio**
Mila, Université de Montréal, CIFAR

## Abstract

GFlowNets are probabilistic models that learn a stochastic policy that sequentially generates compositional structures, such as molecular graphs. They are trained with the objective of sampling such objects with probability proportional to the object's reward. Among GFlowNets, the temperature-conditional GFlowNets represent a family of policies indexed by temperature, and each is associated with the correspondingly tempered reward function. The major benefit of temperature-conditional GFlowNets is the controllability of GFlowNets' exploration and exploitation through adjusting temperature. We propose a *Learning to Scale Logits for temperature-conditional GFlowNets* (LSL-GFN), a novel architectural design that greatly accelerates the training of temperature-conditional GFlowNets. It is based on the idea that previously proposed temperature-conditioning approaches introduced numerical challenges in the training of the deep network because different temperatures may give rise to very different gradient profiles and ideal scales of the policy's logits. We find that the challenge is greatly reduced if a learned function of the temperature is used to scale the policy's logits directly. We empirically show that our strategy dramatically improves the performances of GFlowNets, outperforming other baselines, including reinforcement learning and sampling methods, in terms of discovering diverse modes in multiple biochemical tasks.

## 1 Introduction

Generative Flow Networks (GFlowNets) Bengio et al. [2021], Malkin et al. [2022], Bengio et al. [2023], Madan et al. [2023], Deleu et al. [2022, 2023], Pan et al. [2023b, 2022, 2023a], Zhang et al. [2022a, 2023b,a] offer a training framework for learning generative policies that sequentially construct compositional objects to be sampled according to a given unnormalized density or reward function. Whereas other generative models are trained to imitate a distribution implicitly specified by a training set, GFlowNets's target distribution is specified by a reward function seen as an unnormalized density function. The primary advantage inherent to GFlowNets is their capacity to uncover a multitude of highly rewarded samples from a diverse set of modes Bengio et al. [2021] of the given target distribution. This holds great significance in the context of scientific discovery, exemplified by domains such as drug discovery Bengio et al. [2021], Jain et al. [2022, 2023].

Temperature-conditional GFlowNets are training frameworks for learning conditional generative models with $p(x|\beta)$ proportional to a tempered reward function: $p(x|\beta) \propto R(x)^\beta$. In contrast to typical GFlowNets trained a single distribution with fixed temperature $\beta$, temperature-conditional GFlowNets learn a family of generative policies corresponding to reward functions raised to different

---

[*]equal contribution, correspondence to min-su@kaist.ac.kr

NeurIPS 2023 AI for Science Workshop

powers. The major benefit of temperature-conditional GFlowNet is that $\beta$ is controllable because it conditions the learned generative policy, and we find this to be a crucial factor in managing the exploration and exploitation trade-off.

Temperature-conditional GFlowNets have already been introduced Zhang et al. [2022b], Gagrani et al. [2022] and have shown promising results for specific neural architectures, such as in the case of *Topoformer* Gagrani et al. [2022] for solving scheduling problems Zhang et al. [2022b]. However, training a neural network with different temperature parameters showing up in its loss function can lead to challenges due to the high variance of gradients. Consequently, achieving stable training and consistently powerful performance in applications of temperature-conditional GFlowNets, such as scientific discovery tasks, remains a significant challenge.

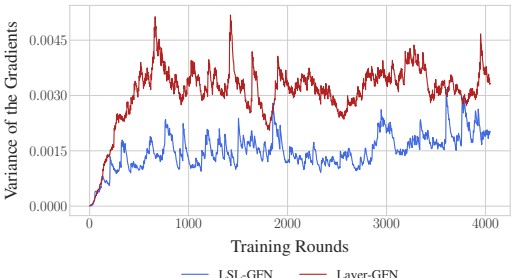

Figure 1: Compared to the Layer-GFN, which represents prior temperature-conditional GFlowNets, our LSL-GFN exhibits a substantially lower variance in parameter gradient, indicating better training stability.

In this paper, we propose *Learning to Scale Logits for temperature-conditional GFlowNet* (LSL-GFN), with an architecture design of the temperature-conditional GFlowNet to obtain a simple yet stable training framework (see Figure 1). Our key idea is to train an auxiliary model which scales the policy's logits directly based on the reward temperature, without interfering with the remaining neural network parameters and usual GFlowNet architecture. We hypothesize that the LSL-GFN, which shares the same neural network across temperatures, helps generalization and results in faster training of temperature-conditional GFlowNets. This innovative approach enhances the efficacy of generalization and exploration within uncharted regions of state space, even when confronted with constraints imposed by limited computational resources in scientific discovery tasks (e.g., drug discovery).

In our empirical findings, the proposed training method, which directly scales logits, outperforms typical approaches that adjust neural network embeddings based on temperature values. The LSL-GFN demonstrates marked enhancements over baseline GFlowNet methodologies, specifically in the context of identifying a greater number of modes under constrained reward computation resources. Moreover, our method exhibits superior performance when compared to alternative benchmarks, including both Reinforcement Learning (RL) Schulman et al. [2017], Haarnoja et al. [2017] and Markov Chain Monte Carlo (MCMC) sampling techniques Xie et al. [2020]. The validity of our concept was thoroughly substantiated across diverse biochemical tasks encompassing DNA sequencing and QM9 molecule optimization.

## 2   Preliminaries

Generative flow networks, known as GFlowNets, constitute a class of deep generative models and reinforcement learning (RL) methodology designed to sample compositional objects $x \in \mathcal{X}$. A generated object corresponds to the *terminal state* in a Markov decision process starting in a unique initial state. The learned GFlowNet policy sequentially modifies the state variable by picking an *action* at each step, which triggers a deterministic transition to the next state. For the most part, GFlowNets have been developed in the context where $x$ and the states and actions are discrete, but see recent work extending to continuous domains [Lahlou et al., 2023].

In this paper, we will focus on discrete GFlowNets, whose generative policy $P_F$ is implemented by a neural network with a softmax output layer following the computation of output logits. GFlowNets have been analyzed with the perspective of particle flows [Bengio et al., 2021, 2023], represented by a learned function $F_\theta$, which specifies the relative probabilities of different trajectories from a source to a sink. These flows correspond to unnormalized densities of state transitions and quantify nodes and edges of a directed acyclic graph (DAG), where directed edges signify transitions between nodes, each of which corresponds to a state. It's noteworthy that the DAG structure (in opposition to a tree structure) offers a pivotal advantage by enabling the modeling of numerous potential action sequences that converge upon identical states. This is different from soft Q-learning and entropy-regularized

RL [Haarnoja et al., 2017, 2018, Buesing et al., 2020], which are closely related to GFlowNets but may misbehave in the DAG setting where there are multiple ways of landing in the same terminal state.

The *initial state*, referred to as $s_0$, serves as the starting point for all trajectories, while the *terminal states*, denoted as the last state $s_n$ of a trajectory $\tau = (s_0, s_1, \cdots, s_n)$, functions as a sink for the flows of particles arising from the initial state. We may denote $s_n = x \in \mathcal{X}$, and the terminal states are a subset of all the possible states. The states are connected through deterministic transitions chosen by the policy. The *children* of state $s$, labeled $\text{Ch}(s)$, are reachable from their parent state $s$ through forward transitions denoted by $s \to s' \in \text{Ch}(s)$. The *parent states* associated with a state $s$ are designated as $\text{Pa}(s)$, and they result from backward transitions represented as $s \dashrightarrow s' \in \text{Pa}(s)$.

A crucial concept is that of a *complete trajectory*, $\tau$, which encompasses a sequence of states $(s_0 \to \cdots \to s_n)$, commencing from the initial state $s_0$ and culminating in a terminal state $s_n$. Each transition $(s_t \to s_{t+1})$ within the trajectory must adhere to the set of valid transitions $\mathcal{A}_{s_t}$. The *trajectory flow*, denoted by $F(\tau)$, is a non-negative function that maps complete trajectories to unnormalized probabilities, illustrating the flow of probability from source to sink along that trajectory.

This flow is further dissected into *state flows* and *edge flows*. The state flow, represented by $F(s)$, encapsulates the unnormalized probability (how many particles) passing through a specific state $s$, computed as the sum of $F(\tau)$ over all trajectories $\tau$ containing state $s$. On the other hand, edge flow, expressed as $F(s \to s')$, characterizes the unnormalized probability (how many particles) traversing the transition from state $s$ to state $s'$.

Policies play a pivotal role in governing transitions. The *forward policy*, $P_F(s_{t+1}|s_t)$, quantifies the probability of transitioning from a state to any of its child states, while the *backward policy*, $P_B(s_t|s_{t+1})$, captures the probability of transitioning from a state to one of its parent states. When $F$ is a *Markovian flow*, the forward policy and backward policy can derived as follows: $P_F(s'|s) = F(s \to s')/F(s)$, and $P_B(s|s') = F(s \to s')/F(s')$.

Eventually, we define the key concept, a marginal distribution, denoted as $P_F^\top(x) := \sum_{\tau \to x} P_F(\tau)$, which aggregates probabilities of trajectories terminating at a specific state $x$. The learning problem of GFlowNet is approximately achieving the following conditions:

$$\pi(x) := P_F^\top(x) = \sum_{\tau \to x} P_F(\tau) = \frac{R(x)}{\sum_{x \in \mathcal{X}} R(x)} \tag{1}$$

Eq. (1) can be satisfied in several ways by introducing constraints that apply to the learned flows or policies. The first proposed [Bengio et al., 2021] such constraint (and the corresponding training objective) is Flow matching (FM):

$$\sum_{s' \in \text{Pa}(s)} F_\theta(s \to s') = \sum_{s'' \in \text{Ch}(s)} F_\theta(s \to s'') \tag{2}$$

The equation represents an intuitively clear consistency condition, establishing that the quantity of input flows should equate to the quantity of output flows.

As FM follows a Temporal Difference (TD) approach (a blend of Monte Carlo and dynamic programming methods), while the Trajectory Balance (TB) method, proposed by Malkin et al. Malkin et al. [2022], functions akin to a Monte Carlo (MC) technique in RL. TB aims to improve credit assignment by focusing on complete trajectories and providing a training signal to every transition based on how well a trajectory-level constraint is satisfied. The core principle of TB mandates that the flow of forward trajectories initiated from the source must match the flow of backward trajectories initiated from the sink:

$$Z_\theta \prod_{t=1}^n P_F(s_{t+1}|s_t; \theta) = R(x) \prod_{t=1}^n P_B(s_t|s_{t+1}; \theta). \tag{3}$$

Here, $Z_\theta$ serves as the estimated flow model through the initial state $s_0$ and it also estimates the partition function, the sum of all trajectory flows: $Z_\theta = \sum_{\tau \in \mathcal{T}} F(\tau) = \sum_{x \in \mathcal{X}} R(x)$ when the constraint is satisfied, i.e., when the corresponding mismatch loss is minimized.

Other loss functions have been proposed to satisfy Eq. (1), such as sub-trajectory balance Madan et al. [2023], guided-trajectory balance Shen et al. [2023] and detailed balance Bengio et al. [2023].

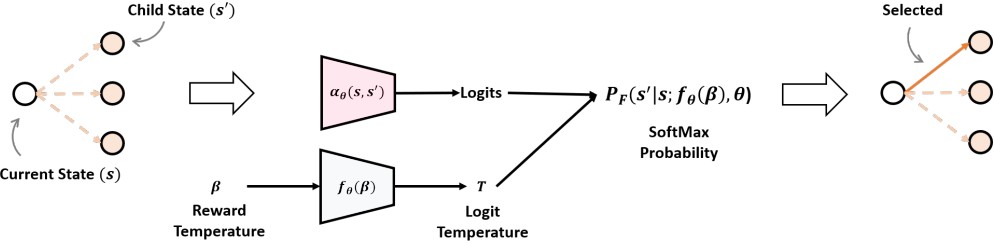

Figure 2: Illustration of the logit-conditional method.

# 3 Temperature-conditional GFlowNets

In this section, we describe two methods for temperature-conditional GFlowNets. Both strategies pursue a common objective: to learn the generative distribution associated with any given temperature (exponent of the reward function). This conditioning is specified by a parameter $\beta$ such that at the convergence of training, we obtain the temperature-conditional generative policy $p(x|\beta) \propto R(x)^{\beta}$.

First, we introduce *layer-conditioning*, which follows earlier temperature-conditional GFlowNet methodologies initially developed by Zhang et al. [2022b] to address scheduling problems. The *layer-conditioning* approach involves the adjustment of neural network parameters based on variations in the temperature parameter $\beta$.

As for the second method, we introduce an innovative architecture known as LSL-GFN. This architecture retains consistent neural network parameters across different temperature values $\beta$ while directly scaling logits regarding the temperature values $\beta$.

## 3.1 Layer-conditioning: Layerwise Concatenation of Temperature Embedding for Conditional Distribution

A conventional approach for constructing a conditional model involves concatenating the conditioning values directly into a model layer [Song et al., 2020, Ho et al., 2020, Zhang et al., 2022b]. Building upon this paradigm, we introduce a *layer-conditioned* GFlowNet (Layer-GFN) that integrates temperature embeddings directly into the model parameterized by $\theta$. Our aim is to train $p(x|\beta) \propto R(x)^{\beta}$ across various $\beta \sim P(\beta)$. We therefore define a mapping $g_{\theta} : \mathbb{R} \to \mathbb{R}^d$, with $d$ representing the hidden dimension size of our model. This $d$ dimensional embedding $e \in \mathbb{R}^d$, represented as $e = g_{\theta}(\beta)$, is incorporated into the model which parameterizes the flows and policies, e.g., concatenated at the initial layer of a graph neural network (GNN).

Moreover, the function, $\alpha_{\theta} : S \times S \times \mathbb{R}^d \to \mathbb{R}$ incorporates the value of $g_{\theta}(\beta)$ when computing the logit by appending these values to the input of the model. Thus, the forward policy in this conditional model is described as:

$$P_F(s'|s; g_{\theta}(\beta), \theta) := \frac{\exp(\alpha_{\theta}(s, s', g_{\theta}(\beta)))}{\sum_{s'' \in Ch(s)} \exp(\alpha_{\theta}(s, s'', g_{\theta}(\beta)))} \qquad (4)$$

## 3.2 LSL-GFN: Conditionally Adjusting the SoftMax Logit Temperature

While layer-conditioning is a useful method for constructing a temperature-conditional GFlowNet, introducing temperature inputs to the model, which parameterizes the flows and policies, may not be sufficient to make sure temperature is properly taken into account. In particular, our experiments suggest that training a temperature-conditional GFlowNet is numerically more difficult than training an unconditional GFlowNet for each temperature. To address this and take advantage of the potential of sharing parameters across temperatures (and generalize across temperatures), we introduce a novel and efficient temperature-conditioning architectural feature termed LSL-GFN. The objective of this method aligns with that of layer-conditioning: to facilitate the training of temperature-conditional GFlowNet $p(x|\beta) \propto R(x)^{\beta}$ over varying temperatures $\beta \sim P(\beta)$. Unlike the ordinary layer-conditioning, we introduce a simple skip connection to adjust the softmax temperature $T$ of logits of $P_F$ as a function of $\beta$, $\alpha_{\theta} : S \times S \to \mathbb{R}$. To be specific, the forward policy $P_F(s'|s; \theta)$ can be

**Algorithm 1** Scientific Discovery with Temperature-Conditional GFlowNets

---
1:  Set $\mathcal{D} \leftarrow \emptyset$                                                      ▷ *Initialize dataset.*
2:  **for** $t = 1, \ldots, T$ **do**                                                          ▷ *Training $T$ rounds*
3:      $\beta_1, \ldots, \beta_M \sim P_{\exp}(\beta)$                          ▷ *Sample temperatures from exploration query prior.*
4:      **for** $m = 1, \ldots, M$ **do**
5:          $\tau_m \sim P_F(\tau | \beta = \beta_m; \theta)$                    ▷ *Sample trajectory from Temp-GFN.*
6:          $\mathcal{D} \leftarrow \mathcal{D} \cup \{\tau_m\}$
7:      **end for**
8:      **for** $k = 1, \ldots K$ **do**                                                 ▷ *Training $K$ epochs per each training rounds*
9:          Use ADAM for gradually minimizing $\mathcal{L}(\theta; \mathcal{D})$.
10:     **end for**
11: **end for**
12: Output: $\mathcal{D}$

---

defined as follows:

$$P_F(s'|s; T, \theta) := \frac{\exp(\alpha_\theta(s, s')/T)}{\sum_{s'' \in Ch(s)} \exp(\alpha_\theta(s, s'')/T)} \tag{5}$$

The logit temperature $T$ determines the confidence level of $P_F$ as a lower $T$ makes a narrow decision, and a higher $T$ gives a smooth decision. By adjusting the scalar value $T$ as a function of $\beta$, we can adjust the output generative distribution easily without much parameterization.

Thus, we define a simple learnable scalar-to-scalar function $f_\theta : \mathbb{R} \to \mathbb{R}$, which turns conditioning variable $\beta$ into softmax logit temperature $T$. The hope is that there exists maps $f_\theta$ which make forward policy $P_F(s'|s; f_\theta(\beta), \theta)$ approximately generate samples $x \sim p(x|\beta) = \prod_{t=1}^{n-1} P_F(s_t|s_{t-1}; f_\theta(\beta), \theta)$ proportional to the tempered reward: $p(x|\beta) \propto R^\beta(x)$.

### 3.3 Training Objective

Our training procedure is based on the trajectory balance (TB) loss, so we aim to minimize $L_{TB}(\tau)$ given a training replay buffer or dataset $\mathcal{D}$ similar to prior GFlowNet work. The difference is that we train the GFlowNets with multiple values of $\beta \sim P(\beta) = U^{[1, 2.5]}$ where $U$ is uniform distribution as used in the loss function:

$$\mathcal{L}(\theta; \mathcal{D}) := \mathbb{E}_{P(\beta)} \mathbb{E}_{P_\mathcal{D}(\tau)} \left[ \left( \log \frac{Z_\theta(\beta) \prod_{t=1}^{n} P_F(s_t|s_{t-1}; f_\theta(\beta), \theta)}{R^\beta(x) \prod_{t=1}^{n} P_B(s_{t-1}|s_t; f_\theta(\beta), \theta)} \right)^2 \right]. \tag{6}$$

When considering the parameterization of deep neural networks (DNNs) using $\theta$, a key implementation feature lies in the conditioning of the partition function $Z$ on the input temperature $\beta$, i.e., we write $Z_\theta(\beta)$ as a learned function rather than a learned constant. Additionally, the conditional dependencies of $P_F$ and $P_B$ are established by the DNN $f_\theta$. This architecture necessitates the incorporation of two auxiliary DNNs, namely $Z_\theta$ and $f_\theta$; however, it's worth noting that these mappings operate from a scalar to a scalar, mitigating the need for an excessive number of parameters. Note this training objective for conditional GFlowNets was originally suggested by Bengio et al. [2023].

**More epochs.** We apply the ADAM optimizer Kingma and Ba [2014], to minimize the loss function $\mathcal{L}(\theta; \mathcal{D})$. Among the crucial hyperparameters, the number of epochs applied to the replay buffer or dataset $\mathcal{D}$ holds a significant role. A low number of epochs before sampling new trajectories can result in underfitting, while an excessive number of epochs may lead to overfitting. In contrast to the unconditional GFlowNet, we observed that temperature-conditional GFlowNets require more epochs at each round to adapt to various temperature conditions effectively. Consequently, we allocate a fourfold increase in the number of epochs for training temperature-conditional GFlowNets when comparing them with their unconditional counterparts; see Figure 5b for detailed analysis.

### 3.4 Scientific Discovery Algorithm with Temperature Conditional GFlowNet

In scientific discovery (e.g., molecule optimization), we discover multiple objects $x \in \mathcal{X}$ so that generative candidate dataset $\mathcal{D} = \{\tau_1, \ldots, \tau_N\}$ (where each trajectory finalized to corresponding objects $\tau \to x$). What people care about $\mathcal{D}$ is not only the average of TopK rewards among $\mathcal{D}$, but also diversity and the number of modes (a local peak in which reward is above the threshold).

Typically, GFlowNet training involves an exploratory policy that autonomously generates trajectories that are then used to perform gradient updates on the GFlowNet parameters. We can significantly enhance the exploratory phase of GFlowNet by querying multiple values of $\beta$ with a temperature-conditional GFlowNet. This enables the model to generate a diverse set of candidates sampled from various generative distributions:

$$\mathcal{D} \leftarrow \mathcal{D} \cup \{\tau_1, \ldots, \tau_M\}, \quad \tau_1, \ldots, \tau_M \sim \int_\beta P_F(\tau|\beta) dP_{\exp}(\beta). \tag{7}$$

The temperature prior $P_{\exp}(\beta)$ controls the range of temperatures for exploration, and in our experiments, we set $P_{\exp}(\beta) = U^{[1,3]}$. See Algorithm 1 for detailed pseudo code.

## 4 Experiments

In this section, we present our experiment results on two biochemical tasks, QM9 and TFBind8. QM9 is a molecule optimization task that tries to maximize the HOMO-LUMO gap from density functional theory Zhang et al. [2020], and TFBind8 is a DNA sequence design task whose reward function is wet-lab measured binding activity to a transcription factor. Following a recent work Shen et al. [2023], we formulate our problems as a sequence prepend/append MDP, where there is more than one trajectory per $x$. See detailed implementation and hyperparameters in Appendix A.

**Baselines.** To evaluate the proposed methods, we first compare them with the prior representative GFlowNet method, TB Malkin et al. [2022]. We also consider MARS Xie et al. [2020], which is an MCMC-based method known to work well in molecule domain, and RL-based methods which include A2C with Entropy Regularization Mnih et al. [2016], Soft Q-Learning Haarnoja et al. [2018], and PPO Schulman et al. [2017] as baselines.

**Evaluation Metric.** In biochemical tasks, the diversity of generated samples is particularly important due to the uncertainty of the oracle itself Bengio et al. [2021]. To this end, our primary objective is to discover as many modes (unique dissimilar samples above a certain reward level) as possible; the number of modes metric is most crucial. Then, we also mean reward and diversity of high-scoring 100 samples (Top-100) among those 2000 samples generated from the model every 10 training rounds for monitoring performances.

### 4.1 Performance Evaluation in QM9

The objective of the QM9 task is to construct a molecular graph using a predetermined set of 12 building blocks, each comprising 2 stems. This results in the creation of a molecule composed of 5 such blocks, adhering to the configuration outlined by Shen et al. [2023]. The primary aim here is to maximize the HOMO-LUMO gap, a critical molecular property, which is derived from a pre-trained MXMNET model as a proxy, as outlined in [Zhang et al., 2020]. To appreciate the complexity of this problem, one must consider both the solution space $\mathcal{X}$ and the trajectory space $\mathcal{T}$, which are described as follows: ($|\mathcal{X}| = 58,765, |\mathcal{T}| = 940,240$).

In the context of the QM9 tasks, temperature-conditional GFlowNet methods stand out as they consistently outperform all the baselines, which include TB, RL methods, and sampling techniques, particularly in terms of the number of modes while the LSL-GFN outperforms the layer-conditioning method. When considering the Top-100 reward criterion, temperature-conditional GFlowNets attain comparable yet slightly superior performance to TB, surpassing the other baseline methods. Moreover, in the case of Top-100 diversity, LSL-GFN achieves a level of diversity similar to that of TB, while the layer-conditioning method gives less diverse samples. We attribute the success of both LSL-GFN and layer conditioning methods to our new techniques, which involve increasing the number of training epochs, denoted as $K$, and fine-tuning exploration using the temperature prior $P_{\exp}(\beta)$ alongside the corresponding algorithm presented in Algorithm 1. When comparing LSL-GFN and

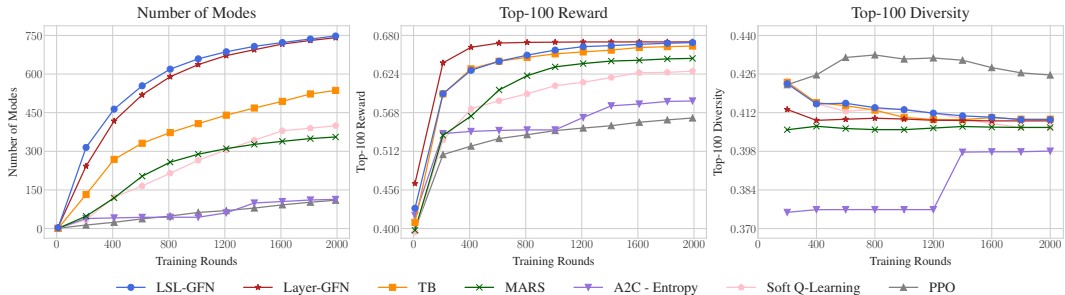

Figure 3: Performance evaluation in QM9. All experiments are conducted on three random seeds.

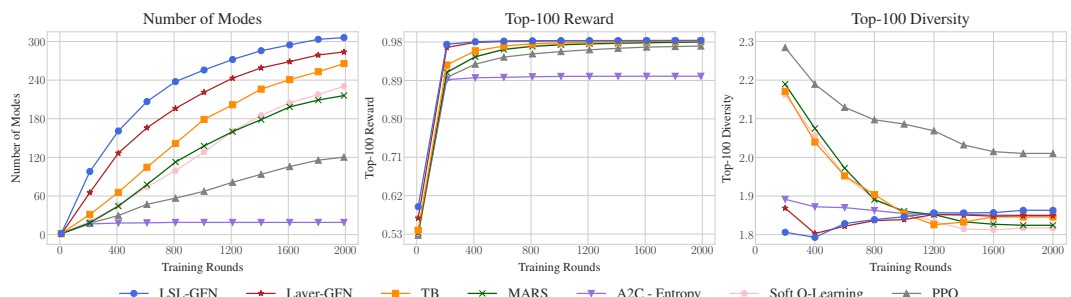

Figure 4: Performance evaluation in TFbind8. All experiments are conducted on three random seeds.

layer-conditioning, we find that LSL-GFN exhibits a slightly better number of modes. However, this difference is relatively small, given the simplicity of the QM9 task, making it easy to train both temperature-conditional GFlowNets.

## 4.2 Performance Evaluation in TFbind8

The TFbind8 task aims to generate a nucleotide string of length 8. While conventional practice employs an autoregressive Markov Decision Process (MDP) for string generation, we opt for a prepend/append MDP approach. The reward in this task is determined by the DNA binding affinity to a human transcription factor, as detailed in Trabucco et al. [2022]. The intricacies of the solution space $\mathcal{X}$ and trajectory space $\mathcal{T}$ are as follows: ($|\mathcal{X}| = 65, 536, |\mathcal{T}| = 8, 388, 608$).

We employ a Markov decision process (MDP) to generate DNA sequences for the TFbind8 task using the prepend-append strategy, allowing us to place sequence tokens either at the beginning (prepend) or the end (append) of a partial sequence, as described in Shen et al. [2023]. As a result, multiple action trajectories remain available, all of which can represent the same DNA sequence.

Within the context of the prepend-append MDP setting, it is evident that RL methods exhibit subpar performance across all metrics. This underperformance can be attributed to an exploration strategy that doesn't account for symmetries, resulting in complex exploration where multiple trajectories, denoted as $\tau$, can lead to the same solution, represented as $x$. In contrast, GFlowNet methods excel in this scenario by taking into consideration the symmetries inherent in the problem through the utilization of learned flow models over Directed Acyclic Graph (DAG) structures.

Remarkably, among the various GFlowNet methods, it is the temperature-conditional GFlowNets using our innovative algorithm Algorithm 1 that consistently outperform the others, demonstrating their superior performance. Within temperature-conditional GFlowNets, our new architecture consistently delivers superior performance compared to the layer conditioning method. Given that TFbind8 involves a more complex trajectory space (with $|\mathcal{T}| = 8, 388, 608$) compared to the QM9 task (with $|\mathcal{T}| = 940, 240$), these results highlight the potential of LSL-GFN to provide more stable training than layer conditioning method as the task complexity increases.

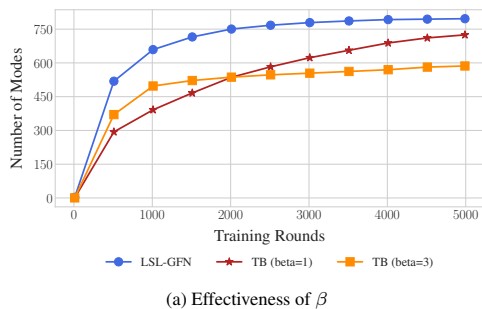
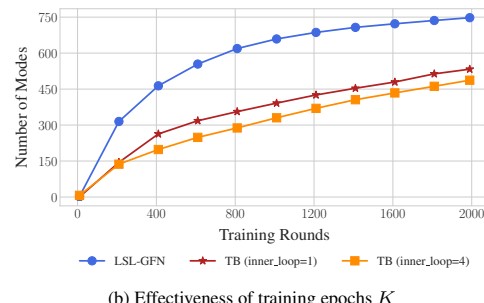

(a) Effectiveness of $\beta$          (b) Effectiveness of training epochs $K$

Figure 5: Evaluating the impact of temperature parameter $\beta$ and training epochs on the performance of temperature-conditional GFlowNet.

### 4.3 Ablation Study

There can be several research questions for the ablation study. We conducted two ablation experiments on the QM9 task to clarify the two research questions below:

**Q1.** Temperatrue-conditional GFlowNet learns from several temperatures of $\beta$. This includes both low and high temperatures. What if we increase $\beta$ for training TB?

**A1.** To validate the effectiveness of temperature-conditional GFlowNet, we conducted experiments on QM9. We made experiments on several fixed $\beta$ from low to high temperatures to train TB and report the resulting trends in the Figure 5a. We find that increasing $\beta$ does not guarantee performance improvements for TB. However, when we train our method, LSL-GFN, with low to high temperatures and query high temperatures when conditioning and computing losses and gradients, we can achieve a remarkable performance gain.

**Q2.** Temperature-conditional GFlowNet includes more training epochs than TB using the same dataset $\mathcal{D}$. What if we increase the training epochs for TB?

**A2.** Increasing training epochs make extremely decreasing exploration and diversity for unconditioned TB, presumably due to the cause of overfitting. But we could achieve higher performance for our method, LSL-GFN, by increasing the number of training epochs as shown in the Figure 5b.

## 5 Conclusion

We introduced the *Learning to Scale Logits for temperature-conditional GFlowNets* (LSL-GFN), a novel architecture enhancing the training of temperature-conditional GFlowNets. This approach addresses numerical challenges in deep network training brought about by diverse gradient profiles from varying temperatures. By adopting a temperature-learned function to scale the policy's logits, we significantly alleviate these challenges. Empirical evaluations confirm that LSL-GFN surpasses benchmarks such as reinforcement learning in diverse biochemical tasks.

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

# A  Detailed Experimental Setting

## A.1  Implementation

In our GFlowNet implementations, we adhere closely to the methodologies outlined in [Shen et al., 2023]. We take the approach of re-implementing only those methods that do not already exist in the literature. All of our GFlowNet models incorporate prioritized replay training (PRT) and the parametrization mapping of relative edge flow policy (SSR), as originally proposed by Shen et al. [2023]. To encode both forward and backward policies, we employ a multi-layer perceptron (MLP) that accepts one-hot encodings of states as input. Following prior research, we introduce a reward exponent denoted as $\beta$ in the reward function, such that $\pi(x) \propto R(x)^{\beta}$.

For the specific GFlowNet policy model, we implement an MLP architecture with relative edge flow parameterization (SSR), as recommended in Shen et al. [2023]. When dealing with pairs of states $(s, s')$, we encode each state into a one-hot encoding vector and concatenate them as input for the forward and backward policy networks. For QM9 tasks, we employ a two-layer architecture with 1024 hidden units, while for the TFbind8 task, we opt for a two-layer architecture with 128 hidden units. Both forward and backward policies use the same architecture but with different parameters. We initialize $\log Z_{\theta}$ to 5.0 for the baseline implementation.

To facilitate temperature conditioning parameterization, transitioning from $\beta$ to $T$, we employ a two-layer MLP with a 32-dimensional hidden layer and Leaky ReLU activation. Similarly, for parameterizing $Z_{\theta}(\beta)$, we use a two-layer MLP with a 32-dimensional hidden layer and Leaky ReLU, mirroring the architecture of $f_{\theta}$.

## A.2  Hyperparameters

Regarding the hyperparameters for GFlowNets, we maintain the initial settings proposed by Shen et al. [2023] without alteration. Across all tasks, we employ the Adam optimizer [Kingma and Ba, 2014] with the following learning rates: $1 \times 10^{-2}$ for $Z_{\theta}$ and $1 \times 10^{-4}$ for both the forward and backward policy. Furthermore, we adopt distinct reward exponents and reward normalization constants as recommended by Shen et al. [2023]. For the inner looping training hyperparameter, we set $K = 4$, and $K = 1$ for baseline GFlowNet methods. For every method, we set the number of training rounds $T = 2,000$. We set temperature prior for exploration as $P_{\text{ext}}(\beta) = U^{[1,3]}$ and temperature prior for training as $P(\beta) = U^{[1,2.5]}$. It's important to note that we have established the default reward as $R(x)^5$ for the QM9 task and $R(x)^3$ for the TFbind8 task, in accordance with the methodology outlined in [Shen et al., 2023]. The reward exponent $\beta$ is subsequently applied to modify the default reward, resulting in $R(x)^{5\beta}$ for the QM9 task and $R(x)^{3\beta}$ for the TFbind8 task.

In the implementation of RL baselines, we utilize the same MLP architecture as employed in the GFlowNet baselines. The optimization of hyperparameters is achieved through a grid search approach on the QM9 task, with a focus on determining the optimal number of modes. For the A2C algorithm with entropy regularization, we segregate parameters for the actor and critic networks. The selected learning rate of $1 \times 10^{-4}$ is chosen from a range of options, including $\{1 \times 10^{-5}, 1 \times 10^{-4}, 1 \times 10^{-4}, 5 \times 10^{-3}, 1 \times 10^{-3}\}$, and we incorporate an entropy regularization coefficient of $1 \times 10^{-2}$ selected from $\{1 \times 10^{-4}, 1 \times 10^{-3}, 1 \times 10^{-2}\}$.

In the case of Soft Q-Learning, we opt for a learning rate of $1 \times 10^{-4}$ selected from the same set of values: $\{1 \times 10^{-5}, 1 \times 10^{-4}, 1 \times 10^{-4}, 5 \times 10^{-3}, 1 \times 10^{-3}\}$. For the PPO algorithm, we introduce an entropy regularization term of $1 \times 10^{-2}$ and employ a learning rate of $1 \times 10^{-4}$, similarly chosen from $\{1 \times 10^{-5}, 1 \times 10^{-4}, 1 \times 10^{-4}, 5 \times 10^{-3}, 1 \times 10^{-3}\}$. The entropy regularization coefficient is selected from $\{1 \times 10^{-4}, 1 \times 10^{-3}, 1 \times 10^{-2}\}$.

