# OpenReview forum: "Learning to Scale Logits for Temperature-Conditional GFlowNets"
_NeurIPS.cc/2023/Workshop/AI4Science — NeurIPS2023-AI4Science Poster_

### Official Review · Reviewer_pwbq · 2023-10-14
**Interesting paper requiring additional experimental details**

**Rating:** 7
**Confidence:** 4

**Review:**

Temperature scaling [1] is a standard technique to calibrate the outputs of deep probabilistic models and a well-established approach to bias the samples of deep generative models toward higher- or lower-likelihood regions of the modelled distribution [2]. In this paper, the authors apply this approach to GFlowNets by rescaling the logits of the forward policy network and show that sampling trajectories with higher temperatures (i.e. more diffuse reward functions) leads to an enrichment of unique samples over a certain reward threshold.

While I believe that some clarifications are needed regarding the experimental evaluation of the proposed methods, I think this is an interesting and mostly sound paper that would be a good addition to the workshop.

---

## Strengths:
- The paper provides a well-structured introduction and background on the underlying methodology (GFlowNets) and the proposed extensions (Layer-GFN and LSL-GFN)
- The authors carry out an extensive empirical comparison of Layer-GFN and LSL-GFN with five baseline methods across two different tasks.
- An ablation study is performed, highlighting the benefit of training a model over a mixture of temperature scales

## Weaknesses:
- While I agree that increased mode coverage is desirable in many scientific contexts, the *number of modes* metric only reflects the number of unique samples above a certain reward threshold - which may well belong the the same mode. I think that renaming it to "number of unique samples" would be a more accurate reflection of what it measures. It would also be helpful so specify how the threshold was determined.
- It seems that the Top-100 Diversity may be a more appropriate metric to measure the diversity of the generated samples, however, the paper doesn't mention how it is calculated or if larger or smaller values are better. I think this needs further clarification in the main text.
- Similarly, I think the ablation over the number of epochs per training round could be explained a bit better. The paper doesn't explain what the *inner_loop* parameter means, so it is difficult to say whether setting it to four is a meaningful comparison.

## Suggestions:
- It would be interesting to see what the mapping of $f_\theta:\beta\to T$ looks like. As both are scalar, this could be included as a plot, potentially overlaying the learned functions at different training epochs.

---

## References:

[1] Guo, Chuan, et al. "On Calibration of Modern Neural Networks." International Conference on Machine Learning. PMLR, 2017.

[2] Liang, Percy, et al. "Holistic evaluation of language models." Transactions on Machine Learning Research (2022).

---

### Official Review · Reviewer_LBfA · 2023-10-24

**Rating:** 7
**Confidence:** 3

**Review:**

**Summary**

GFlowNets are promising models trained to sample with probability proportional to the reward. This property is especially useful in the drug discovery domain where one is often interested in generating a diverse set of high reward molecules. A quintessential feature of molecular generation is balancing exploration and exploitation which can be balanced in GFlowNets by tuning a temperature parameter. However, training such models can experience numerical instability. To address this, the authors introduce two approaches to train temperature-conditioned GFlowNets: Layer-GFN which concatenates temperature embeddings during training and LSL-GFN which introduces a learnable temperature function and enables parameter sharing across temperatures. Empirically, both approaches outperform previous methods on the QM9 and TFBind8 tasks.

**Main Review**

The paper is well written and easy to follow. The problem is well motivated as balancing exploration and exploitation is ubiquitous in molecular generation. The proposed approach offers advantages in decreasing the variance of gradients during training and empirically shows improved performance.

**Comments**

* What is the reward threshold in the number of modes metric? It would be informative to visualise the distribution of rewards generated by each method. In an extreme case, if the reward threshold is 0.5 and a method generates 10 unique examples passing this, then the number of modes found is 10. But if another method finds 1000 unique examples with 0.49 reward, this would be more desired in certain scenarios.

* How is the reward computed for QM9? The Layer-GFN curve plateaus. Are higher rewarding molecules generated?

* How permissive is the TFBind8 evaluation? The reward saturates quickly for Layer-GFN and LSL-GFN. The number of modes continues to increase indicating new nucleotides are found but do these involve a relatively large number of base edits? If TFBind8 yields high reward in a permissive way, an algorithm which finds a few examples of high reward sequences will quickly be able to generate more. If this is the case, running additional seeds may see the performance difference not as pronounced.

* How is the diversity calculated? It looks like the scale is different on the QM9 and TFBind8 plots.

Overall, the new approach to train temperature-conditioned GFlowNets will be useful to balance exploration and exploitation. In addition, the proposed LSL-GFN architecture stabilises the training gradients' variance which is valuable for any application.

---

### Meta-Review · Area_Chair_Dghg · 2023-10-27

**Recommendation:** Accept (Poster)
**Confidence:** 5

**Metareview:**

&nbsp;

I'm delighted to recommend acceptance of the paper to the workshop! Both reviewers are to be commended for providing extensive feedback on the paper and I encourage the authors to take into account all the points raised.

&nbsp;

Additionally, in future work, the authors may want to include as a competitor method Maus et. al [1] which uses Bayesian optimisation to accomplish similar goals to GFlowNets. The authors may also be interested in the sample efficiency matters benchmark [2] which assesses a wide range of methods, including GFlowNets on molecule optimisation.

&nbsp;

As a very minor nit, I would refer to QM9 as a physical chemistry dataset rather than a biochemical dataset since the 19 regression values are physical properties of the molecules.

&nbsp;

I greatly look forward to discussing the work in further detail at the workshop!

&nbsp;

__**References**__

&nbsp;

[1] Maus, N., Wu, K., Eriksson, D. and Gardner, J., 2023, April. Discovering Many Diverse Solutions with Bayesian Optimization. In International Conference on Artificial Intelligence and Statistics (pp. 1779-1798). PMLR.

[2] Gao, W., Fu, T., Sun, J. and Coley, C., 2022. Sample efficiency matters: a benchmark for practical molecular optimization. Advances in Neural Information Processing Systems, 35, pp.21342-21357.

&nbsp;